# Torque Generation of the Endodontic Instruments: A Narrative Review

**DOI:** 10.3390/ma15020664

**Published:** 2022-01-17

**Authors:** Sang Won Kwak, Ya Shen, He Liu, Hyeon-Cheol Kim, Markus Haapasalo

**Affiliations:** 1Department of Conservative Dentistry, Dental and Life Science Institute, Dental Research Institute, School of Dentistry, Pusan National University, Yangsan 50612, Korea; endokwak@pusan.ac.kr; 2Department of Oral Biological and Medical Sciences, Division of Endodontics, Faculty of Dentistry, University of British Columbia, Vancouver, BC V6T 1Z3, Canada; yashen@dentistry.ubc.ca (Y.S.); endoliuhe@gmail.com (H.L.); 3Department of Stomatology, Affiliated Hospital of Jining Medical University, Jining 272000, China

**Keywords:** endodontics, nickel-titanium file, root canal shaping, stress, torque

## Abstract

As the use of nickel-titanium (NiTi) file systems for root canal therapy has become popular; hence, knowledge and understanding of the characteristics of NiTi files is essential for dentists. Unintended sudden fracture can occur during root canal shaping, and it is important to understand the conditions that may cause instrument fractures. Torque is defined as the force required to rotate the NiTi file and can be considered of as a parameter for the stress generated. The endodontic engine maintains a constant rotational speed by adjusting torque regardless of the root canal conditions. The process of root canal shaping by rotary instruments is a series of actions that requires torque and generates stress to both the teeth and the NiTi instruments. The generated stress may induce the strain accumulation on NiTi instrument and the canal wall and lead to the development of microcrack in the instrument and dentinal wall. Therefore, understanding of torque and stress generated is important to prevent the fractures to the instrument and the teeth. This stress has been measured using various experimental approaches, including microcrack observation by using a microscope or computed tomography, attaching strain gauges to the teeth, and finite element analysis. This review focuses on the stress generated to the teeth and the instrument during instrumentation under various experimental conditions. The factors related to torque generation are also discussed.

## 1. Introduction

Since nickel-titanium (NiTi) material was introduced to endodontic industry for manufacturing files in 1988, NiTi alloys have undergone metallurgical improvements [1]. NiTi instruments offer better elastic flexibility in bending and twisting, as well as superior mechanical properties, when compared to conventional stainless steel files [2,3]. However, unpredictable fracture that may occur during instrumentation of root canals still remains a concern [4]. 

The failure mechanisms of NiTi instruments have been thoroughly investigated [5]. Flexural fatigue is induced by recurrent compressive and tensile strains at the site of maximal flexure when the instrument rotates and moves freely up and down in a curved canal. Torsional failure takes place when the instrument (usually tip) engages and binds within the canal, but the remainder of the file continues to rotate, generating stresses beyond the ultimate strength of the material [5,6]. A fractured fragment of an instrument affects the treatment prognosis, especially when infection is present, because it makes further cleaning and disinfection of the canal difficult [7]. Removal of fractured file from root canal can be challenging and time-consuming, with documented success rates ranging from 55% to 80% [8,9]. Removal of additional dentin by special burs and ultrasonic tips during the removal of the fragment may weaken the root. Therefore, despite success in fragment removal in many cases, prevention of file fracture is important.

Endodontic hand files can be used to clean and shape root canals. The tactile feedback that the clinician receives by holding the hand file is a major advantage of employing hand files for cleaning and shaping. The use of hand NiTi files such as ProTaper Universal hand files (Dentsply Maillefer, Ballaigues, Switzerland) and Profile ISO 04 hand files (Dentsply Maillefer) reduces procedural errors, because they are more flexible than stainless steel files [10]. NiTi files also can be used with a handpiece in a rotary motion to prepare root canals because of their superior flexibility and the design of the cutting blades. Endodontic engines designed specifically for NiTi rotary instruments have been developed to maintain the same rotational speed by adjusting the torque during the root canal shaping. Studies have demonstrated that rotary instrumentation is superior to manual instrumentation in terms of shaping ability and efficiency [11,12]. However, the use of endodontic engine for the canal shaping requires higher torque and faster rotating speed setting. 

Despite the recent advancements in metallurgy and manufacturing technologies that have allowed manufacturing instruments that are more flexible and resistant to fracture, the lifespan of the NiTi instrument is also related to its stress level, and the instrument failure might occur as a result of inappropriate use. Torque is one of the factors that influence the frequency of instrument failure. The engine driven rotary NiTi instrument generates the torque when the file starts rotating [13]. During the root canal shaping, the rotary NiTi instrument are under continuous and variable stress condition depending on the root canal shape and dimensions [14]. A wireless technology (X-Smart IQ; Dentsply Maillefer) was introduced to show the performance of NiTi rotary instruments in vivo by using a software that indicate the instant torque level during root canal preparation. By using a special endodontic motor, the real-time torque generated to each file over time can be monitored [15].

To date, many studies have been conducted based on cyclic fatigue and torsional resistance to evaluate the mechanical properties of the NiTi instrument. However, the root canal shaping is performing under dynamic condition, and these static methods seem to have limitations in reflecting the clinical situation. Therefore, this review focuses on the importance of torque in endodontics and the factors related to dynamic torque generation under various experimental conditions.

## 2. Meaning of Torque in Endodontics

In mechanics, torque is defined as a measure of the force required to rotate an object around an axis [16]; it is also known as moment or moment of force. In the same way that force causes an object to accelerate in linear kinematics, torque causes an object to acquire angular acceleration [17]. Likewise, the meaning of torque in endodontics can be defined as the force that requires to rotate the NiTi file. When using an engine-driven NiTi file, the endodontic engine must rotate at a constant rotational speed, regardless of the root canal conditions such as calcification or degree of root canal curvature, to maintain an acceptable stress condition for the NiTi file [18]. Therefore, by adjusting torque, which is a force applied to a file, the endodontic engine maintains a constant speed of rotating file.

During the instrumentation, the torque generated is influenced by the condition of the root canal or the multi-directional force given by the operator [19]. The size of the preoperative root canal volume is related to the width of the contact area between the root canal walls and the cutting blades of the NiTi file [20]. In proportion to the area of the contacts, the amount of torque generated during the root canal shaping will be affected. The root canal dentin is shaped by the instrument’s reaction force during preparation. The interactions between the NiTi instrument and the canal walls during the preparation may cause momentary stress concentrations in dentin when the file operates inside the root canal [21]. By applying a light force to the root canal during the preparation, the amount of torque generated can be reduced.

The experimental approach for determining the torsional resistance of NiTi files is to apply a single rotational load until the file fractures. The torsional load can be defined as torque around the shaft [22]. Many studies have examined the torsional resistance of the NiTi file by locking 3 mm from the tip while the motor rotates at 2 rpm till the file fracture occurs, which is based on International Organization for Standardization (ISO) 3630-1 (Figure 1a) [23]. However, clinically this monotonic torsional load until the file fracture is not applied during the root canal treatment due to the auto-reverse function in endodontic motor. Reciprocation NiTi file systems are designed to repeat clockwise and counterclockwise movements at certain angles during root canal preparation, which means the reciprocation movement is not limited only to one direction. Therefore, correlating laboratory findings to clinical conditions should be considered with caution and clinically reliable measurement method for comparing the torsional resistance is still required [24].

## 3. Stress Induction to Dentin and NiTi File

Root canal shaping with engine-driven NiTi files can weaken the root by removing the root dentin structure and generating the stresses [25,26]. These generated stresses may induce the strain accumulation on the canal wall and lead to the development of a crack, especially near the apical portion of the instrument [27,28,29]. Because stresses in the canal wall are difficult to evaluate experimentally, there have been some approaches to analyze the stress generated during root canal shaping; (i) by counting the number of crack lines before and after the root canal preparation, (ii) by attaching strain gauges to the tooth, and (iii) by using finite element analysis (FEA). 

It is still controversial whether root canal preparation causes microcracks in dentin. Several studies have observed the development of structural weaknesses such as microcracks and craze lines in root dentin after root canal preparation with rotary NiTi instruments. Capar et al. [30] reported that canal shaping with heat-treated NiTi rotary instruments tended to cause fewer dentinal cracks compared with the conventional NiTi instrument under a stereomicroscope. Riu et al. [31] also reported that the number of canal manipulations with NiTi files were related to the accumulation of damage and the kinematics of the file system also affects the development of dentinal microcracks using a microscope. On the contrary, several studies by using micro-computed tomography showed that root canal instrumentation did not result in the formation of dentinal microcracks and questioned about the previous findings [32,33]. These differences in experimental results appear to be due to the experimental methods (i.e., using micro–computed tomography or sectioning the tooth), the condition of teeth during the shaping procedure, and the kind of which NiTi file system was used.

It seems impractical to measure the stress and strain on the root canal directly. Strain gauges have been used to quantify the surface strain from specific locations at the tooth and, as a result, to evaluate the effect from loss of tooth structure or root canal shaping (Figure 1b) [28,29]. The advantage of using strain gauge is that the result shows the changes in stress applied to root dentin during the process in real time. However, the strain gauges cannot be placed inside the root canal and only can be used at a few locations on the root surface, which may not reflect strain patterns inside the dentin. Furthermore, the stresses generated by various loads were highest on the canal wall and low on the root surface [34]. The measurement of root surface stresses does not seem to offer the precise internal root canal stresses. 

FEA has been used to evaluate stresses in a variety of study fields. FEA is an engineering method for the numerical analysis of stress distribution and concentration depending on the material properties and loading conditions [35]. The benefits using FEA for endodontics include precise modelling of complicated geometry of instruments or root canals, systematic model change, and analysis of stress patterns induced within the root canal when it is subjected to the force [34]. During root canal preparation, a canal is shaped by the contacts created by instrument and dentin walls. These contacts create momentary stress concentrations in dentin [21]. Contact stress levels are established by the mechanical behavior of NiTi file systems, which is determined by their design. The NiTi file systems with different geometric features generates different levels of circumferential tensile stress during the canal preparation and residual stress [36]. The thickness of residual dentin and external root morphology can also affect the pattern of stress generated during root canal shaping [37].

NiTi file systems are also subjected to various stresses during the root canal shaping. FEA has been used to measure and compare the amount of stress generated to the NiTi file. FEA also provides the information on the site of stress concentration occurring within the NiTi file. The area where the stress is concentrated is one of the most susceptible locations to file fracture and fatigue initiation [36]. Stress generated levels in rotary instruments depend not only on the bending and applied forces but also on instrument properties. Several studies proved that torsional deformation and bending stress were related to the NiTi file’s longitudinal and cross-sectional design [38,39]. Heat treatment can also improve the instrument’s flexibility and help to reduce bending moment and maximum stress value [40,41]. Interestingly, residual strains and stresses can exist when the file is not subjected to the force after the root canal shaping [35], and these invisible stresses might weaken the instrument. Nevertheless, it is often difficult to confirm whether the 3D modeling of NiTi file or root canal was appropriate, without verification. Inaccurate geometric modelling of experimental subjects may lead to inaccurate results.

## 4. Dynamic Torque Generation during Canal Preparation

The torque is one of the parameters that shows stress generated during the root canal preparation. Several studies reported the instrumentation of root canals with the rotary NiTi file caused more crack formation depending on the speed of torque settings [42,43,44]. In the higher torque setting, the instrumentation technique generates more torque, and more reaction force to dentin would be generated during instrumentation. Since Sattapan et al. [45] measured torque generated and the apical force applied during instrumentation under dynamic condition by mounting the torque meter in the tooth, several studies have been carried out to investigate the dynamic torque by using torque sensor and strain gauges [46,47,48,49].

To date, many studies about torsional characteristics of NiTi files have been conducted under a static mode, which does not reflect clinical situation. Root canal shaping is a dynamic procedure. During the instrumentation, the removal of root dentin is accompanied by changes in the curvature angle and size of the root canal. In 1999, Blum et al. [44] reported that the crown-down technique generated lower torque than step-back technique by analyzing the vertical forces and torque generated in the extracted teeth during the root canal preparation. Their device equipped with force transducers allowed to record the torque generated during the preparation. Similarly, there was an attempt to measure the torque generated by collecting the torque data from the endodontic motor directly [13,50,51,52]. Recently improved software techniques enabled the creation of devices capable of evaluating and recording real-time torque during the root canal preparation (Figure 2). With the development of these technologies, it is possible to easily obtain the data of torque change patterns according to file systems or the different experiment conditions of root canal. The studies that evaluated the dynamic torque generated are summarized in Table 1. 

The root canal treatment is performed under very complex circumstances, and the torque generation is determined by various factors. The geometric characteristics and heat treatment of NiTi file system have an effect on the generated torque. Peters and Barbakow [53] reported that the larger diameter instruments necessitate higher apical pressures to penetrate deeper into the root canals. Kwak et al. [51] investigated the cross sectional area of NiTi file had a greater impact on the torque generation than kinetics did. Cross sectional design is related to the contact area between the file system and canal wall, while the longitudinal design such as taper affects volumetric dimension and diameter of the NiTi file. Reducing the contact area between instrument and root canal would decrease the friction and generate the lower torque during root canal shaping [49,50,51]. The heat treatment is related to the flexibility of the instrument. During the instrumentation in a curved root canal, the NiTi file tends to revert to its original shape, which leads to increase of the friction between the file and the canal wall. The flexibility of NiTi files seem to have less effect on torque generation in a straight canal, while the more flexible files such as controlled memory files generated less torque in a curved canal above 35° [13]. 

The kinetics of NiTi file system also affects the torque generated. Kwak et al. [13] observed the different patterns of torque generated between continuous rotary and reciprocation movement (Figure 3). The reciprocation file systems rotate clockwise and counterclockwise direction repeatedly at a specific angle during their movement. When the rotation direction changes from clockwise to counterclockwise, an instantaneous torque is generated. For this reason, the reciprocation file systems generated more torque than continuous rotary file systems in a straight canal. Recently introduced, the adaptive motion includes both continuous and reciprocating motions and switches its motion depending on the workload. This alternative movement for the NiTi file system generated lower torque compared to the continuous motion, especially when the coronal pre-flaring was insufficient [46,51,54]. 

The host factor such as handling behavior can also affect the torque generated. During the root canal preparation, a clinician applies continuous up and down movements to the NiTi file with different pecking speed, pecking depth, and apical force. According to the study of Maki et al. [48], the higher pecking speed generated larger vertical force and torque than the low-speed pecking. The time a NiTi file rotates within the root canal is directly related to the total amount of torque generated. Therefore, the pecking motion with low speed may reduce the momentary maximum torque, but the longer preparation time caused by the slow pecking motion may lead to an increase in total amount of torque generated and the risk of cyclic fatigue fracture [48,55]. The pecking depth also affects the torque generated and shorter pecking depth result in lower screw-in forces [55]. The instrumentation technique was also related to the torque generation, where brushing action required less torque than the pecking motion [52]. 

**Table 1 materials-15-00664-t001:** A summary of the methodology of the studies that evaluated the torque generated in this review.

Study	Type	NiTi File Used	n	Canal Type	Evaluated Parameter
Blum et al., 1999 [44]	In vitro	ProFile	-	Natural teeth	Canal shaping technique, Manipulation
Sattapan et al., 2000 [45]	In vitro	Quantec series 2000	5	Natural teeth	File design, root canal size
Peters and Barbakow, 2002 [53]	In vitro	Vortex	12	Plastic S shaped canal	Rotational speed
Pereira et al., 2013 [47]	In vitro	ProTaper Next	6	Plastic canal	Rotational speed, insertion pattern
Kwak et al., 2018 [50]	In vitro	WaveOne, WaveOne Gold	15	Plastic J shaped canal	Heat treatment and design of file, Glide path establishment
Gambarini et al., 2019 [52]	In vivo	Twisted File	10	Natural teeth	Instrumentation motion
Kwak et al., 2019 [51]	In vitro	K3XF, Twisted File Adaptive	15	Plastic S shaped canal	File design, kinetics
Maki et al., 2019 [48]	In vitro	ProTaper Next	7	Plastic J shaped canal	Up-and-down speed
Htun et al., 2020 [54]	In vitro	HyFlex EDM Glidepath, MANI glidepath	10	Plastic S shaped canal	File design, kinetics
Kimura et al., 2020 [46]	In vitro	EndoWave	7	Plastic straight canal	Kinetics
Lee et al., 2020 [56]	In vitro	HyFlex EDM, OneCurve, Twisted File Adaptive, ProTaper Next	15	Natural teeth	File design, kinetics
Kwak et al., 2021 [13]	In vitro	WaveOne, WaveOne Gold, ProTaper Universal, ProTaper Next	15	Metal J shaped block	Root canal curvature, Heat treatment and design of file, kinematics

Anatomical factors such as pre-operative canal size and curvature angle of root canal are also among the factors related to the torque generated. Shaping a narrow canal with a single file without pre-shaping with smaller files results in a wide contact surface between the file and the dentinal walls, causing substantial stresses on both the file and the root canal walls [28]. Therefore, it is recommended to create a glide path before the final shaping procedure. The increase in the preoperative canal volume by creating the glide path will be beneficial to reduce total amount of torque generated [50]. Kwak et al. [13] observed that the maximum and total amount of torque generated increased as the curvature increased, and the conventional (not heat treated) NiTi file systems were most affected by the curvature angle.

## 5. Limitations and Future of Dynamic Torque Study

Despite the fact that research measuring dynamic torque have been extensively conducted, these studies still have drawbacks. Employing the resin blocks for standardization to measure the torque could be criticized because of a difference in hardness from root dentin. Because of the higher hardness of resin blocks compared with natural teeth, additional torque generated could be expected when shaping the artificial root canal. The hardness of natural teeth may also vary depending on the aging or storage status of the teeth. In addition, there are differences in the shape, canal size, and root canal curvature of natural teeth [56]. The amount of torque generated is largely affected by the size of the preoperative root canal. These variables in natural tooth cannot be controlled and may result in significant variation in the results due to this lack of uniformity. Experiments employing artificial teeth with dentin-like hardness can be considered to equalize the size of the root canal, however due to the limitations of 3D printing technology such as the shrinkage of the resin and limited resolution, it appears that reproducing a narrow root canal with a diameter of 10–15 μm without errors is still very challenging.

Furthermore, like in most in vivo experiments, differences in torque generation due to apically applied force or pecking speed between researchers cannot be overlooked. These errors are unavoidable if these factors are not precisely controlled by computerized machines. For these reasons, importing the laboratory results to clinical practice should be made with careful consideration and further studies are required to overcome these limitations. Gathering a massive amount of data using the in vivo study model would be helpful in evaluating actual stress generated in various circumstances.

## 6. Conclusions

When a new file enters the market, extensive research into torsional and cyclic fatigue resistance of the file is attempted. However, there is relatively little research on torque generation, which properly reflects clinical situations. Torque occurs inevitably when using a rotary NiTi file and can be one of the parameter reflecting stress induced to the NiTi file or tooth. Dynamic torque research offers the advantage of real-time stress generated during the root canal shaping. The precision of new torque measurement methods could provide meaningful information about the clinical performance of various engine-driven NiTi instruments.

Clinically, the torque generation is affected by the NiTi instrument’s stiffness, which is determined by design and alloy type. Additionally, the amount of torque generated depends on anatomical features of the root canal and the clinician’s handling behavior. Therefore, in order to avoid instrument failure from excessive torque, the degree of torque generation should be considered by taking all these factors into account.

## Figures and Tables

**Figure 1 materials-15-00664-f001:**
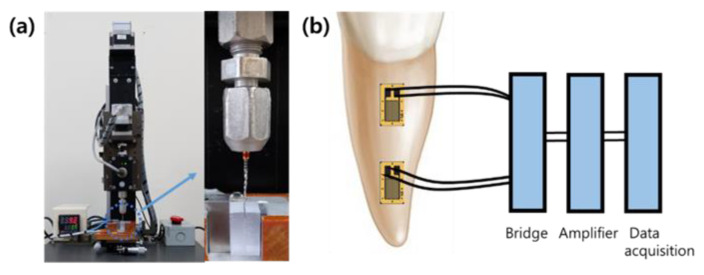
Experimental test set-up to measure the torque and strain (**a**) A custom-made equipment for measuring torsional resistance; (**b**) an experimental setting of attaching the strain–stress gauge to the tooth.

**Figure 2 materials-15-00664-f002:**
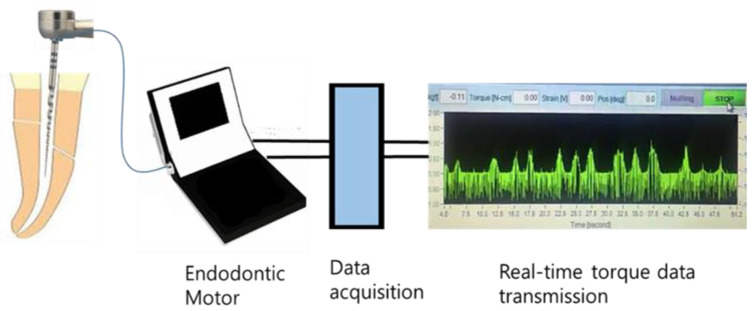
Schematic drawing of experimental set-up for the real-time torque measurement.

**Figure 3 materials-15-00664-f003:**
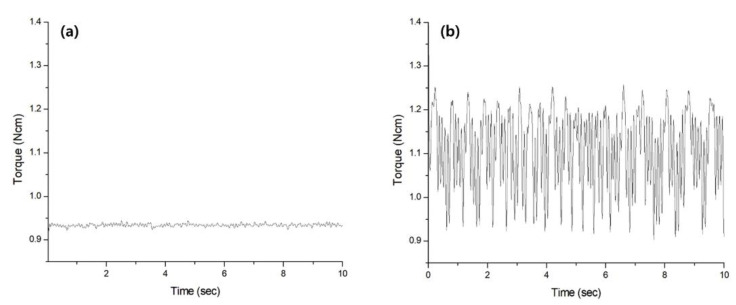
Different torque generation patterns between (**a**) continuous rotary (ProTaper Next; Dentsply) and (**b**) reciprocation movement (WaveOne; Dentsply) in the artificial root canal with 15° angle of curvature [13].

## Data Availability

Data sharing is not applicable to this article.

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
