# Peer review of "Torque Generation of the Endodontic Instruments: A Narrative Review"

_materials, 2022, doi:10.3390/ma15020664_

Round 1

Reviewer 1 Report

Dear Authors, 

You made a great works!

However, some improvements are mandatory!

Author Response

Thank you for reviewing. We have uploaded the revised manusciprt.

Reviewer 2 Report

I would like to congratulate the authors for conducting the present study regarding torque applied for Endodontics instruments.

  • Major concern

A reader after reading this paper becomes completely lost. The authors try to debate torque throwing to the table a few loose ideais without a clear narrative flow. What was the purpose of the paper after all? Some subheadings do not represent very well the text under them. The authors should improve the subheading sequence, improve the narrative speech flow, make it read friendly, and clearly assume an objective of the the review and chase that objective on a comprehensive manner.

  • Here goes a few of my minor concerns:

This seems to be a narrative review. Therefore the term “narrative review” should appear in the title.

E recommend to add the term “Endodontics” to the keywords

The Introduction is not giving any clue regarding the rationale of the review. Although this is not an experimental study, that does not mean it does not need a rationale and a study purpose. I recommend the authors to work a little bit better this part.

In STRESS INDUCTION TO DENTIN It has been proved that micro cracks on dentin are strongly related with the methodology employed. This debated should be updated. The De Deus group have a lot of material related with this topic.

The conclusion do not represent the review body text. Probably because there is nothing to conclude from a narrative review which does not present a clear objective stated.

Several important things are missing and may improve the review:

  • outcomes. There are so many papers assessing the torsional stress applied to the instrument. I recommend a subheading debating outcomes an previously published results
  • The authors debate several points. But not a very relevant issue such as crystallographic arrangement and the impact on the outcomes. I recommend the authors to debate the recent studies DOI: 10.1111/iej.13463 and 10.1111/iej.13529
  • Another issue is related with the instruments design. Professor McSpadden text book has a lot of information that can be applied to this matter.
  • Multi-method approach to assess instruments profile (DOI 10.1016/j.joen.2020.07.016) has been proposed as an approach that can improve the comprehensive knowledge of the multi factor that can influence the outcomes
  • The torsional test as something that does not mimic the clinic and the truer meaning of this should be mentioned

Author Response

(The authors gave the same response as above.)

Round 2

Reviewer 1 Report

Thank You. 

Reviewer 2 Report

Dear auhtors, I have no more concerns.